# Effect of Soy Protein Products on Growth and Metabolism of *Bacillus subtilis*, *Streptococcus lactis*, and *Streptomyces clavuligerus*

**DOI:** 10.3390/foods13101525

**Published:** 2024-05-14

**Authors:** Wei Wen, Miao Hu, Yaxin Gao, Pengfei Zhang, Weimin Meng, Fengxia Zhang, Bei Fan, Fengzhong Wang, Shuying Li

**Affiliations:** 1Institute of Food Science and Technology, Chinese Academy of Agricultural Sciences, No. 2 Yuan Ming Yuan West Road, Beijing 100193, China; wen1624938238@163.com (W.W.); humiao7890@163.com (M.H.); gaoyx1997@163.com (Y.G.); zhangpf240987@163.com (P.Z.); mengweimin2021@163.com (W.M.); yixi2021@1126.com (F.Z.); fanbei517@163.com (B.F.); 2Key Laboratory of Agro-Products Quality and Safety Control in Storage and Transport Process, Ministry of Agriculture and Rural Affairs, Chinese Academy of Agricultural Sciences, Beijing 100193, China

**Keywords:** nitrogen source, soy protein products, fermentation, nattokinase, nisin, clavulanic acid

## Abstract

Microbial nitrogen sources are promising, and soy protein as a plant-based nitrogen source has absolute advantages in creating microbial culture medium in terms of renewability, eco-friendliness, and greater safety. Soy protein is rich in variety due to different extraction technologies and significantly different in the cell growth and metabolism of microorganisms as nitrogen source. Therefore, different soy proteins (soy meal powder, SMP; soy peptone, SP; soy protein concentrate, SPC; soy protein isolate, SPI; and soy protein hydrolysate, SPH) were used as nitrogen sources to culture *Bacillus subtilis*, *Streptococcus lactis*, and *Streptomyces clavuligerus* to evaluate the suitable soy nitrogen sources of the above strains. The results showed that *B. subtilis* had the highest bacteria density in SMP medium; *S. lactis* had the highest bacteria density in SPI medium; and *S. clavuligerus* had the highest PMV in SPI medium. Nattokinase activity was the highest in SP medium; the bacteriostatic effect of nisin was the best in SPI medium; and the clavulanic acid concentration was the highest in SMP medium. Based on analyzing the correlation between the nutritional composition and growth metabolism of the strains, the results indicated that the protein content and amino acid composition were the key factors influencing the cell growth and metabolism of the strains. These findings present a new, high-value application opportunity for soybean protein.

## 1. Introduction

Microorganisms are diverse and widely distributed in nature [1]. Microorganisms are being increasingly studied based on the strong and growing interest in food production technology and drug development. Microorganisms can be used as fermenters in the food industry, such as *Bacillus subtilis*, the starter strain used for natto production [2]. Nattokinase (NK) and other functional food ingredients are produced during the fermentation of natto. NK, as an effective thrombolytic enzyme, has been intensively studied in clinical practice and is likely to become an important functional substance in the prevention of thrombotic cardiovascular diseases [3]. Microorganisms also act as probiotic agents, for example, *Lactobacillus* and *Bifidobacterium*, which regulate the intestinal ecosystem and improve the hosts’ health [4]. *Streptococcus lactis*, as a probiotic, is characterized by the ability to use carbohydrates as substrates to produce lactic acid and nisin. Nisin, as an efficient, non-toxic, safe, and non-side-effect-inducing natural food preservative additive, is added to dairy products, fermented beverages, canned foods, and meat products to control the growth of pathogenic microorganisms such as *Staphylococcus aureus* and *Listeria monocytogenes*, thereby extending the shelf life of foods [5]. Microorganisms are also used as biosynthetic agents in the pharmaceutical industry; for instance, clavulanic acid (CA), an important secondary metabolite produced by *Streptomyces clavuligerus* of the *Actinobacillus* family, is used clinically for the treatment of infectious diseases [6].

Their wide range of applications in the microbiology industry makes it necessary to cultivate these microorganisms. Microorganisms can grow smoothly and produce metabolites with a high potency only when the nutritional conditions are suitable. The five major nutrients required for microbial growth are carbon and nitrogen sources, water, inorganic salt, and growth factors [7]. Nitrogen sources, as essential nutrients, are involved in the synthesis of the protein biomass, nucleotides, and secondary metabolites required for microbial growth and metabolism, as well as in the biosynthesis of high-value products, such as antibiotics and enzymes [8]. Nitrogen sources can be divided into inorganic nitrogen sources and organic nitrogen sources. An organic nitrogen source with more commercialized conventional applications is beef extract. The traditional animal sources of organic nitrogen sources are not often used due to many issues such as epidemic risks and customs and taboos. However, plant sources of nitrogen are safer and less expensive. At present, soy protein is regarded as a high-quality, nutritionally balanced, and easily accessible plant protein resource displaying an excellent amino acid composition [9].

Soybean meal is a by-product of soybean oil extraction from soybeans and has a protein content of about 44.0–53.1%. Due to its high protein content, balanced amino acid composition, and abundant supply, it is a major source of high-quality protein [10]. Soy protein concentrate (SPC) is obtained by removing the carbohydrates and lipids from soybean meal and has a protein content of more than 70% [11]. There are two methods of preparation: (1) Acid leaching, which involves the precipitation of water-insoluble fibers and proteins using isoelectric point precipitation. (2) Alcohol leaching, which comprises dissolving soluble carbohydrates and alcohol-soluble polyphenols, leaving the soy proteins being [12]. SPC is considered to have a high nutritional value because it is rich in essential fatty acids, phospholipids, minerals, and nine essential amino acids, has a balanced amino acid composition similar to that of animal proteins, and has a digestibility of up to 90% [13]. Soy protein isolate (SPI) is obtained by dissolving the proteins from soybean meal at a high pH, removing the insoluble fibers using centrifugation to obtain a supernatant, and reacquiring the proteins at the isoelectric point, with a content of about 85–90% [14]. Soy protein hydrolysate (SPH) is a mixture of soy proteins that have been converted into oligopeptides, polypeptides, and amino acids prepared either using protein-processing processes, such as hydrolysis and a heat treatment, or biological processes, including digestion and microbial fermentation [15]. Soy peptone (SP) is a powdered product prepared from soybean meal using enzymatic hydrolysis, filtration, concentration, and spray drying. SP contains proteins, peptides, amino acids, water-soluble vitamins, and many sugars. Soluble carbohydrates mainly include sucrose, stachyose, raffinose, and verbascose, which function as gene inducers in specific expression systems [16]. Soy protein is rich in nutrients and can support the growth of various microorganisms and cell cultures in the fermentation, medical, and biological industries. At the same time, it is advantageous in terms of renewability, eco-friendliness, and safety.

In this work, different kinds of microorganisms were cultured by using five kinds of soy protein product (SP, soy powder meal (SMP), SPC, SPI, and SPH) to replace the nitrogen source in a medium, including *B. subtilis*, *S. lactis*, and *S. clavuligerus*. The nutritional components of the different soy protein products were detected before three-strain culturing, and the effects of the five kinds of soy protein product as fermented substrates on cell growth and metabolism were compared to determine the optimal nitrogen source for each strain. A further correlation analysis was performed on the nutritional components of the five kinds of soy protein product and the cell growth and metabolism of the three strains. The main purpose of this research includes two parts: one is to explore the effects of five kinds of soy protein product on the cell growth and metabolism of different types of strains, to target three specific strains and clarify their preferred types of soy protein products, and to achieve amino acid balance to meet the personalized needs of cell growth and metabolism, laying a foundation for the subsequent improvement of soy protein products. The other is to help extend the soybean industry chain, enhance the value chain, broaden the application fields of soybean resources, and realize the high value of soybean protein.

## 2. Materials and Methods

### 2.1. Strains and Materials

*B. subtilis* BSNK-5 was identified in our laboratory and stored at −80 °C in a refrigerator with 15% glycerol (*w*/*v*). *S. lactis* (BNCC 190729) and *Micrococcus luteus* (BNCC 102589) were purchased from BeNa Culture Collection. *S. clavuligerus* (ATCC 27064) was purchased from Shanghai Guyan Industrial Co., Ltd. (Shanghai, China). The yeast extract and tryptone were purchased from Oxoid (LP0021, LP0042B, Beijing, China). DeMan Rogosa and Sharpe (MRS) medium was purchased from AOBOX (02-293, Beijing, China). The Nutrient Broth (NB) medium was purchased from Qingdao Hope Bio-Technology Co., Ltd. (HB0108, Qingdao, China). The malt extract was purchased from Solarbio Science & Technology Co., Ltd. (M8130, Beijing, China).

Urokinase and nisin were purchased from Yuanye Bio-Technology Co., Ltd. (9039-53-6, 1414-45-5, Shanghai, China). CA was purchased from the National Institutes for Food and Drug Control (6SNT-6F3D, Beijing, China).

The beef extract (BE) was purchased from AOBOX (01-009, Beijing, China). SP was purchased from Solarbio Science & Technology Co., Ltd. (1008C052, Beijing, China). SMP, SPC, SPI, and SPH were supplied by Shandong Vandefu Co., Ltd. (Shandong, China).

### 2.2. Strain Cultures and Fermentation

All the strains were activated using solid plates, and then their seed liquids were obtained using liquid culturing. The optimal source of soybean-protein-based nitrogen was screened using fermentation medium. Concretely, BSNK-5 was activated on Luria-Bertani (LB) solid plates for 12 h and incubated at 37 °C at 200 rpm for 12 h to obtain a seed culture. The 2% seed culture was inoculated in a 250 mL conical flask containing 50 mL of fermentation medium at 37 °C in an orbital shaker (TS-300 DC, Tiancheng Experimental Instrument Manufacturing Co., Ltd., Shanghai, China) at 180 rpm for 120 h. The fermentation medium, containing 30 g/L of glucose, 18.34 g/L of BE, 0.79 g/L of CaCl_2_, and 1.3 g/L of MgSO_4_·7H_2_O, was used to determine the relevant indicators every 12 h. *S. lactis* was activated on MRS solid plates for 48 h and incubated at 30 °C for 12 h to obtain a seed culture. The 2% seed culture was inoculated into fermentation medium (50 mL) in a 250 mL conical flask containing 20.0 g/L of glucose, 24.45 g/L of BE, 2.0 g/L of NaCl, 0.2 g/L of MgSO_4_·7H_2_O, and 10.0 g/L of KH_2_PO_4_ at 30 °C for 24 h for further index determination every 4 h. *M. luteus* was cultivated on the NB solid plates at 37 °C for 24 h and incubated in the NB broth at 37 °C for 12 h at 200 rpm until approximately 0.3 OD at 600 nm, which was regarded as an indicator organism to detect the bacteriostatic effect of nisin produced by *S. lactis*. *S. clavuligerus* was activated on ISP-2 solid plates, which consisted of 4.0 g/L of yeast extract, 10.0 g/L of malt extract, 4.0 g/L of glucose, and 20.0 g/L of agar at 28 °C for 4 d, and this was inoculated at 28 °C for 72 h at 260 rpm. The 2% seed culture was inoculated into the fermentation medium (50 mL) in a conical flask (250 mL) containing 20.0 g/L of glycerol, 5.0 g/L of glycerol trioleate, 20.0 g/L of dextrin, 27.71 g/L of BE, 21.0 g/L of morpholine propane sulfonic acid (MOPS), 1.0 g/L of K_2_HPO_4_, 0.1 g/L of NaCl, 0.1 g/L of MgCl_2_, 0.1 g/L of CaCl_2_·2H_2_O, 0.005 g/L of ZnSO_4_, and 0.1 g/L of FeSO_4_·7H_2_O at 28 °C for 144 h at 220 rpm for further index determination every 24 h.

SP, SMP, SPC, SPI, and SPH were used as nitrogen sources to replace the BE in the fermentation medium of BSNK-5, *S. lactis*, and *S. clavuligerus* with total nitrogen contents of 2.25 g/L, 3.4 g/L, and 3.0 g/L, respectively.

### 2.3. Analysis of Components of Five Kinds of Soy Protein Product

The total nitrogen content was determined using the Kjeldahl digestion method (YQ205-09, VELP, Usmate Velate, Italy) [17]. The content of amino acids was measured as follows: each sample was constituted in 10 mL of HCl (6 mol/L) and flushed with nitrogen for 1 min to exhaust the air in the tube. Then, these samples were hydrolyzed in an oven at 110 °C for 24 h. The samples were dispensed into a 50 mL volumetric flask, and then cooled to room temperature. After mixing them in a fixed volume, 1 mL of each sample was taken and nitrogen blown until it was dry. The samples were redissolved to 5 mL with 0.02 mol/L of hydrochloric acid, mixed well, and passed through a 0.2 µm filter membrane into a liquid phase vial, and the composition and content of amino acids were analyzed with an automatic amino acid analyzer (UGC-24C, Yousheng United Technology Co., Ltd., Beijing, China).

### 2.4. Analysis of Cell Growth

The characterization of cell growth metrics included the pH and bacteria density. The partial fermentation of the broth was carried out, and the pH was measured with a meter (PHS-3E, Mettler Toledo Co., Ltd., Shanghai, China). The bacteria density of BSNK-5 and *S. lactis* was measured using the plate count method. Serial tenfold dilutions of the cell suspensions of BSNK-5 and *S. lactis* were prepared. Then, a volume of 0.1 mL of an appropriate concentration of each dilution was spread on the LB solid plates. These plates were incubated at 37 °C for 12 h. The viable cell concentration was calculated and is expressed as colony-forming units per milliliter (CFU·mL^−1^). The bacteria density of *S. lactis* was counted using the same method. The plates spread with cell suspension were incubated at 37 °C for 24 h.

The growth of *S. clavuligerus* is represented as the packed mycelium volume (PMV). The PMV was measured according to the method of Lee [18,19]. The samples of the culture broth (10 mL) were centrifuged for 20 min at 12,000 rpm and separated into supernatant fluids and sediments. The volume of sediments cells is expressed as a percentage.

### 2.5. Measurement of Fermented Substrates in Metabolites

The yield of NK activity is expressed as fibrinolytic activity, measured using the fibrin plate method [20]. The determination of nisin occurred as follows: the testing plate was prepared by pouring 25 mL of molten medium, which contained 5 g/L of glucose, 8 g/L of tryptone, 2.5 g/L of yeast extract, 2 g/L of Na_2_HPO_4_, 5 g/L of NaCl, 1.5% agar (cooled to 40–50 °C), and 1.5% Twen-20 at 70 °C. When the temperature dropped to 50 °C, it was pre-mixed with an overnight culture of the indicator organism (OD_600_ = 0.3) on a sterile plate and allowed to solidify on a horizontal plane at 50 °C [21]. Afterwards, holes were punched in each agar layer using a sterilized puncher, and 100 µL of the nisin standard solutions with gradient dilution or sample dilutions were added into the holes. After incubation at 30 °C for 24 h, a standard curve of nisin inhibition zones versus units of nisin standard was drawn by measuring the diameters of the inhibition zones produced by the nisin standard solution. The nisin titer of the samples was calculated from the standard curve. The measurements of all the samples were performed in triplicate using three different plates.

The CA concentration in the fermentation broth was determined spectrophotometrically. A volume of 0.4 mL of each sample and 2.0 mL of imidazole solution (60 g/L and pH 6.8) were mixed. After the reaction at 30 °C for 15 min, the OD of the mixed liquid was measured at 311 nm [22].

### 2.6. Data Analysis

In this research, three parallel tests were performed for each sample, and the results are expressed as mean ± standard deviation (SD). GraphPad Prism 9.0 was used for data analysis and to plot the charts. Heat maps of the correlation analysis were generated in the R environment with heatmap packages.

## 3. Results and Discussion

### 3.1. Component Analysis of Five Kinds of Soy Protein Product

The total nitrogen content and amino acid composition of nitrogen sources are the key factors affecting microbial growth and metabolism. Therefore, to evaluate the potential of SP, SMP, SPC, SPI, and SPH as sources of nitrogen, first, the total nitrogen content, protein concentration, and amino acid composition of the five kinds of soy protein product were analyzed. As shown in Figure 1A, the total nitrogen content and protein concentration of the five kinds of soy protein product were significantly different. The total nitrogen content of each sample was always higher than their protein content, which means that free amino acids existed in each sample. The total nitrogen content and protein concentration of SPI and SPH were both higher than those of the BE, and those of SP, SMP, and SPC were lower than those of the BE. As shown in Figure 1B, the results expressed the composition and content of 17 amino acids in five kinds of soy protein product. In general, soy protein products showed higher contents of Glu, Asp, Leu, and Arg and lower contents of Met and Cys. The contents of Glu (>100 mg/g) in the five kinds of soy protein product were all higher than those in the BE, while the contents of Gly in the BE were much higher than those in all the soy protein products. It can be seen that the five kinds of soy protein product were different in protein type, total content, and purity, which led to significant differences in the amino acid composition and content.

### 3.2. Effect of Five Kinds of Soy Protein Product on Cell Growth and Metabolism

#### 3.2.1. Effect of Five Kinds of Soy Protein Product on Cell Growth and NK Synthesis of BSNK-5

Cell growth was characterized by the pH and bacteria density. As shown in Figure 2A, the pH of the five soy protein products increased and then decreased during the first 60 h of fermentation, while the pH of the SPI and SPH significantly increased in the second 48 h. This increase in the pH could be explained by protein catabolism and amino acid deamination [23]. In general, the pH of the fermentation broth was alkaline. Some studies have shown that *B. Subtilis* has the ability to synthesize protease at a suitable pH of 7.0–7.4 [13]. Therefore, it was essential that the pH of the fermentation medium was adjusted to 7.0 before sterilization. The bacteria density in different fermentation mediums increased significantly in the first 24 h, and then it decreased and stabilized from 24 to 36 h of fermentation. The bacteria density produced by SMP culturing (4.67 × 10^10^ CFU/mL) was slightly lower than that produced by BE culturing. The bacteria densities of the other four soy protein products were similar, between 3.4 × 10^9^ and 7.0 × 10^9^ CFU/mL. The reason for these differences in bacteria density might have been because of the abundant nutrition in SMP compared with that in the other soy protein products. Other studies have found that SMP has more soluble proteins, and the growth of BSNK-5 will first consume the soluble proteins [23]. Similarly, an appropriate carbon/nitrogen ratio has a great influence on the growth of *B. Subtilis*. The bacteria density of SMP was significantly higher than that of SPC, and the removal of carbohydrates would lead to the loss of nutrients in BSNK-5 growth [13]. In addition to essential nutrients, some metal ions, such as Ca^2+^, K^+^, and Mn^2+^, promote the formation of bacteria and spores [24].

According to Figure 2C, NK activity increased in the first 36 h of fermentation and stabilized in the subsequent fermentation process. The NK activity level of SP was the highest at 36 h, reaching 3.2 × 10^4^ IU/mL, which was 20% higher than that of the BE and 28–60% higher than that of the other four soy protein products. This may have been because BSNK-5 was screened out from the natto food, and soybean originally adds natto bacteria for fermentation. Due to the comparable compositions between SP and natto, it is clear that this strain is more adapted to the environment and is conducive to the production of NK. These data are similar to those achieved in studies that found that the best nitrogen source is SP [25,26]. Based on the comprehensive evaluation of cell growth and metabolic synthesis, SP is the optimal nitrogen source for BSNK-5 fermentation.

#### 3.2.2. Effect of Five Kinds of Soy Protein Product on Cell Growth and Nisin Synthesis of *S. lactis*

In the study, the pH of the fermentation medium, the bacteria density, and the nisin content were determined during fermentation. As shown in Figure 3A, due to the lactic acid produced by *S. lactis* in the fermentation process, the pH of the fermentation broth decreased continuously in the first 16 h of fermentation and stabilized at 16–24 h. When SPC was used as the fermentation nitrogen source, the pH was significantly higher than that of the other soy protein products. As shown in Figure 3B, the *S. lactis* strain produced the highest bacteria density (2.40 × 10^10^ CFU/mL) when BE was used as the fermentation nitrogen source at 20 h. Among the five kinds of soy protein product, the bacteria density produced by SPI was the highest at 8 h. It was found that SPC as a nitrogen source for fermentation was not conducive to cell growth and the metabolite synthesis of *S. lactis*. The reason for this was the loss of the soluble components of SPC during alcoholic extraction [14]. The amounts of nisin produced by SPI at 8 h and SP at 20 h were higher than that of the BE at 12 h. The change in nisin content was consistent with that of bacteria density. Other studies have shown that there is a positive correlation between nisin production level and bacteria density [27]. In addition, the decrease in nisin production can also be explained by acidification caused by lactic acid production [28]. The inhibition of nisin synthesis was because of bacterial autolysis or the release of proteases due to the decrease in the pH in the fermentation broth [29,30,31]. There were three main reasons for the decrease in nisin titer in the later stage of fermentation: (i) Nisin in the culture medium was adsorbed on the producing strain at the later stage of fermentation. (ii) Some proteases produced by the strain may continuously degrade the nisin. (iii) Nisin itself is unstable under neutral conditions and will degrade or inactivate [21,32].

#### 3.2.3. Effect of Five Kinds of Soy Protein Product on Cell Growth and CA Synthesis of *S. clavuligerus*

As shown in Figure 4A, the pH of the fermentation broth was lowest at 72 h, but then it increased. At the end of fermentation, the pHs of all kinds of fermentation broth were alkaline, the pHs of SMP and SPH were the highest, the pH of SPI was in the middle, and the pHs of SP, SPC, and the BE were the lowest. The pH of the medium affected the growth of *S. clavuligerus*, and it increased in the range of 5.0–8.5 [1]. The variations in PMV were complex, with an overall upward trend. The PMV of *S. clavuligerus* rose in the first 120 h of fermentation, and then descended for 120–144 h. The PMV of SPI reached a maximum of about 50% at 120 h, but the PMV of the other four soy protein products was only 25–35%.

As shown in Figure 4C, the content of CA increased in the first 72 h and then stabilized from 72 h to 144 h. CA had the highest stability at a neutral pH, and the decomposition rate was significantly higher at acidic and alkaline pHs [33]. The content of CA produced by SMP was the highest among the soy protein products, followed by SP at 72 h and SPC and SPI at 96 h. Other studies have shown that soy powder and its derivatives are the best nitrogen sources for the production of CA via fermentation, which provides Arg as a 5C precursor for CA synthesis [34]. It has been reported that the addition of the protein extract of soybean to a medium can significantly increase the yield of CA. Compared with the addition of corn steep liquor, when soybean protein extract is used as the sole nitrogen source, the yield of CA is three times higher than that of rice steep liquor [1].

### 3.3. Dissecting the Correlation between the Soy Protein Products and Strains in Growth and Metabolites

As shown in Figure 5, blue indicates a positive correlation, while red indicates the opposite. A positive correlation indicates a promoting effect, and a negative correlation indicates an inhibiting effect. Figure 5 reveals the correlation between the nutritional components of the soy protein products, the cell growth of BSNK-5, and NK synthesis. It has been reported that the pH influences bacterial growth and enzyme synthesis [35]. There was a negative correlation between the pH and bacteria density. NK was synthesized by BSNK-5, and the fermentation broth environment was weakly alkaline, which made the activity level of NK activity as an alkaline serine enzyme higher. Since BSNK-5 utilized glucose to produce organic acids, resulting in a decrease in the pH of the fermentation broth, the correlation analysis validated a negative correlation between the pH and NK [23]. At 36 h of fermentation, the bacteria density was positively correlated with the peptides and Pro. In addition, the NK activity was positively correlated with the peptides and Pro. The bacteria density was positively correlated with NK. Other studies have shown that NK synthesis consumes energy, and the energy source is mainly amino acids. The higher the degree of the hydrolysis of soybean protein is, the higher the concentrations of peptides and amino acids are. NK synthesis consumes peptides, resulting in an increase in NK production [36].

Figure 6 indicates the nutritional components of the soy protein products, the cell growth of *S. lactis*, and nisin synthesis. The bacteria density was positively correlated with the protein content, but when the substrate concentration was too high, the growth of the bacteria was inhibited. The bacteria density was negatively correlated with nisin. With the growth of *S. lactis*, nisin was continuously produced. The reason for the negative correlation between the bacteria density and nisin might have been that the strain entered a stable period and nisin was no longer produced. The production of nisin was heavily dependent on the biomass of the strain, and high-density strains promoted the production of nisin. Therefore, many studies aim to increase nisin production by increasing the physiological activity of bacteria or the number of active nisin-producing bacteria during fermentation. One of the most basic and effective methods is to increase the bacteria density [21]. The bacteria density was negatively correlated with the pH, and a lower pH inhibited the growth of bacteria. Nisin was positively correlated with the pH, and the accumulation of lactic acid led to a decrease in the pH, which, in turn, inhibited the growth of *S. lactis*. Nisin is highly stable at a low pH. With an increase in the pH, the stability of nisin decreases greatly [37]. After heating at 121 °C for 15 min, the nisin activity level decreased by 29% at pH 4.0, 69% at pH 5.0, and 86% at pH 6.0. When the pH changed to 7.0, 99.7% of the nisin lost its antibacterial activity [38]. Nisin was positively correlated with Gly, Ala, Met, Glu, Ser, and Pro. Other studies have shown that adding the precursor amino acids of nisin (Cys, Thr, and Ser) and sulfur-containing amino acids (Met and Cys) to a basic medium can increase the titer of nisin in the fermentation broth [37]. Therefore, the concentration of nisin can be increased by modulating the pH environment and amino acid composition of the medium.

The correlation between the nutritional components of soy protein products, the cell growth of *S. clavuligerus*, and CA synthesis are shown in Figure 7. The protein content was closely related to the cell growth and CA content of *S. clavuligerus*. PMV was positively correlated with the protein content. PMV was positively correlated with Leu, Val, Ile, Asp, and Glu. PMV was negatively correlated with Lys, Arg, Gly, Ala, and Pro. CA was positively correlated with the protein content. CA was positively correlated with Met, Pro, Lys, Gly, Ala, and Arg. Some experiments have shown that the addition of certain amino acids (Thr, Glu, and Asp) does not aid or even inhibit CA synthesis [39]. Complex mediums that provide large amounts of hydrolyzed or free amino acids are more favorable for CA production than chemically defined mediums [39]. For example, Lys, Val, Tyr, Phe, and Cys have strong inhibitory effects [40]. Rodrigues found that, by adding different concentrations of Arg, Thr, Orn, and Glu to a fermentation medium in which soy isolate proteins were used as a nitrogen source, Arg and Thr had no effect on CA production, while the addition of the other two amino acids inhibited CA production [22]. The current findings on CA synthesis are inconsistent, with the addition of Arg and Thr favoring CA synthesis [41], which may be due to the inconsistency of the strains. In response to these results, the CA concentration could be increased by the addition of amino acid supplements to provide C-5 precursors for the urea cycle [39].

## 4. Conclusions

Based on the rapid growth of the microbiology industry and high-quality nutritional components of soy protein products, it is necessary to promote cell growth and metabolism at a low cost in order to develop widely applicable medium nitrogen sources. The total nitrogen contents and protein concentrations of the five kinds of soy protein product were significantly different, and the total nitrogen content of each sample was always higher than its protein content. The total nitrogen contents and protein concentrations of SPI and SPH were both higher than those of SP, SMP, and SPC. The contents of 17 amino acids in the five kinds of soy protein product were ordered as follows: SPC > SPI > SMP > SPH > SP. Among them, the contents of Glu, Asp, Leu, and Arg were higher, and the contents of Met and Cys were lower. BSNK-5 had the highest bacteria density in SMP medium and had the highest NK activity level in SP medium. *S. lactis* had the highest bacteria density in SPI medium and nisin had the highest inhibitory activity level. *S. clavuligerus* had the highest PMV in SPI medium and had the highest CA content in SMP medium. The correlation analysis of the nutritional composition in fermentation broth, the cell growth, and the metabolism of the strains were closely related to the protein content and amino acids. The bacteria densities of BSNK-5 and NK activity were positively correlated with the peptide content. The bacteria density of *S. lactis* was positively correlated with the protein content. Nisin was positively correlated with certain amino acids. The PMV of *S. clavuligerus* and the CA content were both positively correlated with the protein content. These findings suggest that soy protein products have the potential to be novel microbial nitrogen sources.

## Figures and Tables

**Figure 1 foods-13-01525-f001:**
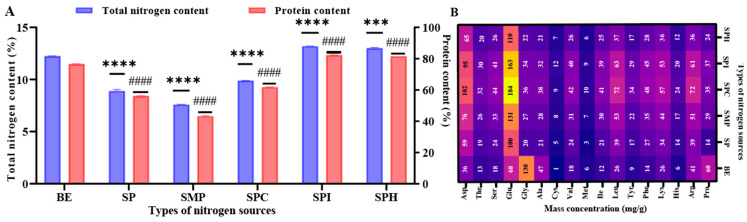
The protein content (**A**) and amino acids analysis (**B**) during the five kinds of soy protein product. Control: BE as the nitrogen source for positive control. *** Significance code: *p* < 0.001; **** Significance code: *p* < 0.0001; and #### Significance code: *p* < 0.0001. Mass concentration (mg/g): mg of amino acids per g of the samples.

**Figure 2 foods-13-01525-f002:**
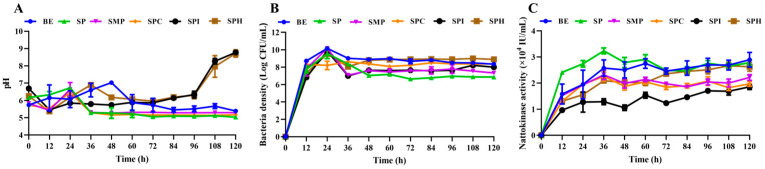
The pH (**A**), bacteria density (**B**), and NK activity (**C**) during the fermentation of five kinds of soy protein product.

**Figure 3 foods-13-01525-f003:**
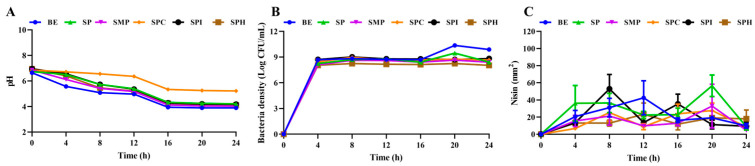
The pH (**A**), bacteria density (**B**), and nisin (**C**) during the fermentation of five kinds of soy protein product.

**Figure 4 foods-13-01525-f004:**
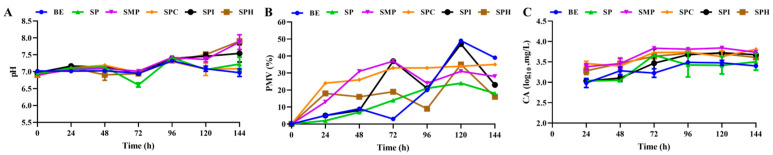
The pH (**A**), PMV (**B**), and CA (**C**) during the fermentation of five kinds of soy protein product.

**Figure 5 foods-13-01525-f005:**
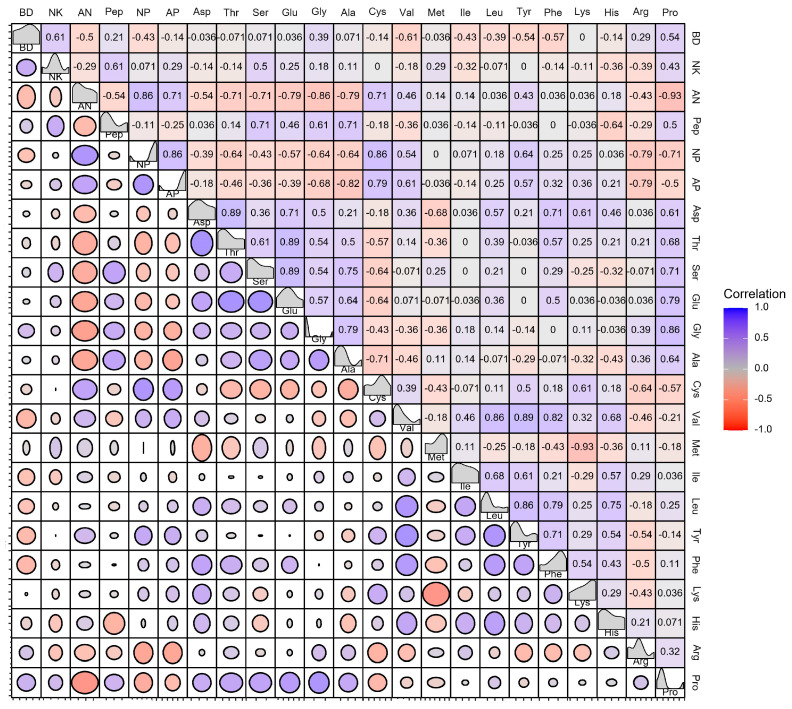
Pearson correlation coefficients between different indexes in BSNK-5 (AN: amino nitrogen; Pep: peptide; NP: neutral protease; AP: alkaline protease; and BD: bacteria density).

**Figure 6 foods-13-01525-f006:**
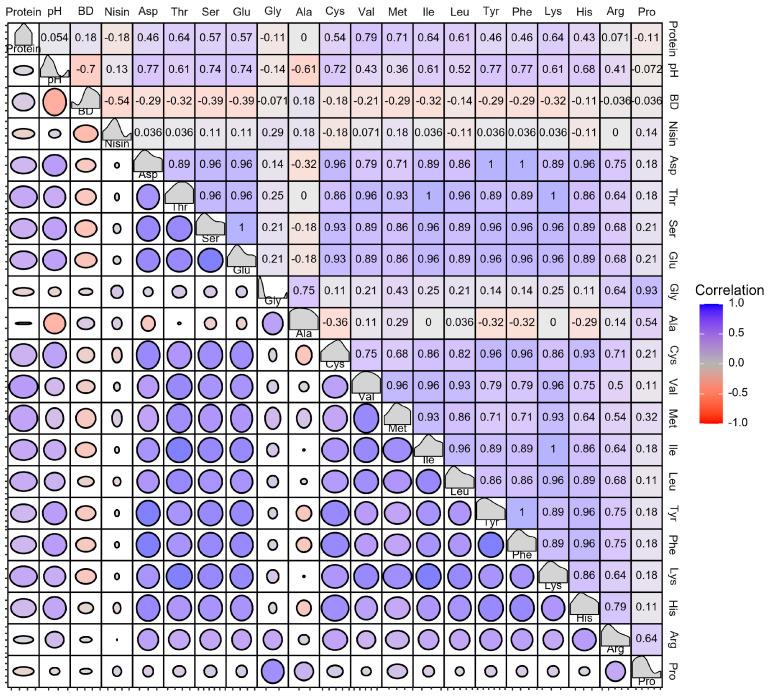
Pearson correlation coefficients between different indexes in *S. lactis*. (BD: bacteria density).

**Figure 7 foods-13-01525-f007:**
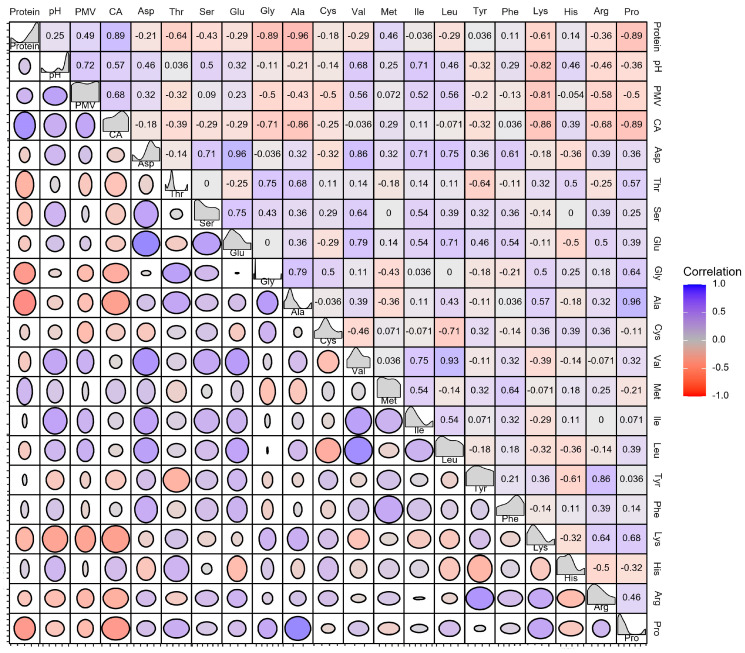
Pearson correlation coefficients between different indexes in *S. clavuligerus*.

## Data Availability

The original contributions presented in the study are included in the article, further inquiries can be directed to the corresponding authors.

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
