# Peer review of "Effect of Soy Protein Products on Growth and Metabolism of Bacillus subtilis, Streptococcus lactis, and Streptomyces clavuligerus"

_foods, 2024, doi:10.3390/foods13101525_

Round 1
Reviewer 1 Report
Comments and Suggestions for Authors
The article entitled “Effect of Soy Protein Products on The Growth and Metabolism of Bacillus subtilis, Streptococcus lactis and Streptomyces clavuligerus” presents interesting results. However, some adjustments to the document are necessary.
Abstract
L(16). In the sentence: “Then they were used as nitrogen sources to culture Bacillus subtilis, Streptococcus lactis and Streptomyces clavuligerus”.
- What is the importance of cultivating this type of microorganisms? for what purpose? Suggest to briefly detail.
L(17). I suggest specifying which metabolites are important to produce and why.
Introduction
L(60 - 62). That idea has another context. I suggest rephrasing it to give better closure to the context of the benefits and properties of soy protein.
L(77-79). This idea could be integrated to the previous one, since it can be understood (from the context provided) that the use of soy protein is not only for its favorable characteristics for the growth of microorganisms as a source of nitrogen, but also for its valorization as a raw material. Thus, it remains as a single objective generates.
Materials and methods
2.1. Strains and Materials
L(86). In the sentence: “MRS medium”.
- Enter the full name of the medium and place the abbreviation in parentheses.
L(87). In the sentence: “NB”.
- Enter the full name of the medium and place the abbreviation in parentheses.
2.2. Strains culture and fermentation
L(99). In the sentence: “LB solid plates”.
- Enter the full name of the medium and place the abbreviation in parentheses.
- Indicate this in section 2.1. Strains and materials.
L(105). Proper spelling of the scientific name (S. lactis).
L(109). In the sentence: “was NB. M. luteus”
- To avoid redundancy with the next sentence, place the name in parentheses.
3. Results and Discussion
3.1. Component analysis of five kinds of soy protein products
L(192). In the Figure 1.B
- Specify the amino acid concentration units. Is it mg per mg of Protein?
3.2.1. Effect of five kinds of soy protein products on cell growth and NK synthesis of BSNK-5
L(205). Proper spelling of the scientific name (B. subtilis).
L(214 – 215). In the sentence: “The reason for the difference in bacteria density might be associated with the abundant nutrition in SMP comparing with other soy protein products. The data of bacteria density indicated that the growth preference of BSNK-5 was SMP”.
- More scientific evidence is needed to support this assertion. At the very least, it is suggested that the concept of "nutrition" be further analyzed. For example, to associate it with the concentration of proteins and the types of amino acids already analyzed above and to corroborate whether or not this has a significant influence. In addition, look for background information or reports that support this statement.
L(227). In the sentence: “The NK produced by B. subtilis has potential in the treatment of cardiovascular diseases”
- This information is relevant; however, it can go in the introduction so that it better justifies the work.
L(230). In the Figure 2.B
- It is suggested to express CFU/mL values in another numerical scale (Ln or Log10). In addition, verify if the data are complete since the graph shows a cut-off.
- In general. Set up the graphs so that the variables tested can be differentiated. The curves appear very mixed between them and make a proper analysis difficult for the reader.
3.2.2. Effect of five kin ds of soy protein products on cell growth and nisin synthesis of S. lactis
L(233). Proper spelling of the scientific name (S. lactis).
L(257). In the Figure 3.B
- It is suggested to express CFU/mL values in another numerical scale (Ln or Log10). In addition, verify if the data are complete since the graph shows a cut-off.
- In general. Set up the graphs so that the variables tested can be differentiated. The curves appear very mixed between them and make a proper analysis difficult for the reader.
3.2.3. Effect of five kinds of soy protein products on cell growth and CA synthesis of S. clavuligerus
L(260). Proper spelling of the scientific name (S. clavuligerus).
L(281). In the Figure 4
- In general. Set up the graphs so that the variables tested can be differentiated. The curves appear very mixed between them and make a proper analysis difficult for the reader.
4. Conclusions
In conclusion, according to the results shown, I consider that a soy protein source suitable for the growth of the 3 microorganisms worked on should be chosen, depending on the biomass production yield or cell density and the metabolites of each one, or failing that, choose one for each microorganism. Likewise, it is also worth mentioning to show in this section, the most outstanding quantitative values obtained according to their objectives.
Comments on the Quality of English Language
The article entitled “Effect of Soy Protein Products on The Growth and Metabolism 2 of Bacillus subtilis, Streptococcus lactis and Streptomyces clavuligerus” presents interesting results. However, some adjustments to the document are necessary.
Abstract
L(16). In the sentence: “Then they were used as nitrogen sources to culture Bacillus subtilis, Streptococcus lactis and Streptomyces clavuligerus”.
- What is the importance of cultivating this type of microorganisms? for what purpose? Suggest to briefly detail.
L(17). I suggest specifying which metabolites are important to produce and why.
Introduction
L(60 - 62). That idea has another context. I suggest rephrasing it to give better closure to the context of the benefits and properties of soy protein.
L(77-79). This idea could be integrated to the previous one, since it can be understood (from the context provided) that the use of soy protein is not only for its favorable characteristics for the growth of microorganisms as a source of nitrogen, but also for its valorization as a raw material. Thus, it remains as a single objective generates.
Materials and methods
2.1. Strains and Materials
L(86). In the sentence: “MRS medium”.
- Enter the full name of the medium and place the abbreviation in parentheses.
L(87). In the sentence: “NB”.
- Enter the full name of the medium and place the abbreviation in parentheses.
2.2. Strains culture and fermentation
L(99). In the sentence: “LB solid plates”.
- Enter the full name of the medium and place the abbreviation in parentheses.
- Indicate this in section 2.1. Strains and materials.
L(105). Proper spelling of the scientific name (S. lactis).
L(109). In the sentence: “was NB. M. luteus”
- To avoid redundancy with the next sentence, place the name in parentheses.
3. Results and Discussion
3.1. Component analysis of five kinds of soy protein products
L(192). In the Figure 1.B
- Specify the amino acid concentration units. Is it mg per mg of Protein?
3.2.1. Effect of five kinds of soy protein products on cell growth and NK synthesis of BSNK-5
L(205). Proper spelling of the scientific name (B. subtilis).
L(214 – 215). In the sentence: “The reason for the difference in bacteria density might be associated with the abundant nutrition in SMP comparing with other soy protein products. The data of bacteria density indicated that the growth preference of BSNK-5 was SMP”.
- More scientific evidence is needed to support this assertion. At the very least, it is suggested that the concept of "nutrition" be further analyzed. For example, to associate it with the concentration of proteins and the types of amino acids already analyzed above and to corroborate whether or not this has a significant influence. In addition, look for background information or reports that support this statement.
L(227). In the sentence: “The NK produced by B. subtilis has potential in the treatment of cardiovascular diseases”
- This information is relevant; however, it can go in the introduction so that it better justifies the work.
L(230). In the Figure 2.B
- It is suggested to express CFU/mL values in another numerical scale (Ln or Log10). In addition, verify if the data are complete since the graph shows a cut-off.
- In general. Set up the graphs so that the variables tested can be differentiated. The curves appear very mixed between them and make a proper analysis difficult for the reader.
3.2.2. Effect of five kin ds of soy protein products on cell growth and nisin synthesis of S. lactis
L(233). Proper spelling of the scientific name (S. lactis).
L(257). In the Figure 3.B
- It is suggested to express CFU/mL values in another numerical scale (Ln or Log10). In addition, verify if the data are complete since the graph shows a cut-off.
- In general. Set up the graphs so that the variables tested can be differentiated. The curves appear very mixed between them and make a proper analysis difficult for the reader.
3.2.3. Effect of five kinds of soy protein products on cell growth and CA synthesis of S. clavuligerus
L(260). Proper spelling of the scientific name (S. clavuligerus).
L(281). In the Figure 4
- In general. Set up the graphs so that the variables tested can be differentiated. The curves appear very mixed between them and make a proper analysis difficult for the reader.
4. Conclusions
In conclusion, according to the results shown, I consider that a soy protein source suitable for the growth of the 3 microorganisms worked on should be chosen, depending on the biomass production yield or cell density and the metabolites of each one, or failing that, choose one for each microorganism. Likewise, it is also worth mentioning to show in this section, the most outstanding quantitative values obtained according to their objectives.
Author Response
Reviewer #1:
Comment 1. L(16). In the sentence: “Then they were used as nitrogen sources to culture Bacillus subtilis, Streptococcus lactis and Streptomyces clavuligerus”. What is the importance of cultivating this type of microorganisms? for what purpose? Suggest to briefly detail.
L(17). I suggest specifying which metabolites are important to produce and why.
Author Response: Thanks for your nice comments. And we apologize to you for the unclearly presentation. We have carefully added the importance of cultivating different types of microorganisms and specified which metabolites are important to produce in the revised manuscript as follows:
L(17-25)The differences in the genetic background of these three strains lead to the expression of different biological enzyme systems and, in turn, different nitrogen requirements during growth and the accumulation of metabolites. Nattokinase (NK), as a characteristic metabolite produced by B. subtilis, reduces hypertensivity, improves blood circulation and reduces the risk of cardiovascular disease. Nisin produced by S. lactis is a natural non-toxic peptide with strong antimicrobial properties against Gram-positive bacteria, which can be used to ensure the freshness and safety of food and prolong its shelf-life. Clavulanic acid (CA), as the main secondary metabolite produced by S. clavuligerus, is used in the clinical treatment of pneumonia and infectious diseases.
Comment 2. L(60-62). That idea has another context. I suggest rephrasing it to give better closure to the context of the benefits and properties of soy protein.
L(77-79). This idea could be integrated to the previous one, since it can be understood (from the context provided) that the use of soy protein is not only for its favorable characteristics for the growth of microorganisms as a source of nitrogen, but also for its valorization as a raw material. Thus, it remains as a single objective generates.
Author Response: We appreciate you very much for your comment. We have rephrased it to give better closure to the context of the benefits and properties of soy protein as follows:
L(68-72)The traditional animal sources of organic nitrogen sources are not often used due to many issues such as epidemic risks and customs and taboos. However, plant sources of nitrogen are safer and less expensive. At present, soy protein is regarded as a high-quality, nutritionally balanced and easily accessible plant protein resource dis-playing an excellent amino acid composition.
Comment 3. L(86). In the sentence: “MRS medium”. Enter the full name of the medium and place the abbreviation in parentheses.
Author Response: Thanks for your comment. We have added the full name of the medium and place the abbreviation in parentheses in the sentence of “MRS solid plates” as follows:
L(118-119)DeMan Rogosa and Sharpe (MRS) medium was purchased from AOBOX (02-293, Beijing, China).
Comment 4. L(87). In the sentence: “NB”. Enter the full name of the medium and place the abbreviation in parentheses.
Author Response: Thanks for your comment. We have added the full name of the medium and place the abbreviation in parentheses in the sentence of “NB solid plates” as follows:
L(119-121)The Nutrient Broth (NB) medium was purchased from Qingdao Hope Bio-Technology Co., Ltd (HB0108, Qingdao, China).
Comment 5. L(99). In the sentence: “LB solid plates”. Enter the full name of the medium and place the abbreviation in parentheses. Indicate this in section 2.1. Strains and materials.
Author Response: Thanks for your comment. We have added the full name of the medium and place the abbreviation in parentheses in the sentence of “LB solid plates” as follows:
L(132-134)Concretely, BSNK-5 was activated on Luria-Bertani (LB) solid plates for 12 h and incubated at 37°C at 200 rpm for 12 h to obtain a seed culture.
Comment 6. L(105). Proper spelling of the scientific name (S. lactis).
Author Response: Thanks for your comment. We have spelt properly the scientific name of S. lactis as follows:
L(138-139)S. lactis was activated on MRS solid plates for 48 h and incubated at 30°C for 12 h to obtain a seed culture.
Comment 7. L(109). In the sentence: “was NB. M. luteus”. To avoid redundancy with the next sentence, place the name in parentheses.
Author Response: Thanks for your comment. We have deleted the “The activation medium of M. luteus was NB” to avoid redundancy with the next sentence as follows:
L(142-145)M. luteus was cultivated on the NB solid plates at 37℃ for 24 h and incubated in the NB broth at 37℃ for 12 h at 200 rpm until approximately 0.3 OD at 600 nm, which was regarded as an indicator organism to detect bacteriostatic effect of nisin produced by S. lactis.
Comment 8. L(192). In the Figure 1B. Specify the amino acid concentration units. Is it mg per mg of protein?
Author Response: The amino acid concentration units in the Figure 1.B indicated per mg of samples as follows:
L(228-229)Mass concentration (mg/g): Amino acid content per mg of samples.
Comment 9. L(205). Proper spelling of the scientific name (B. subtilis).
Author Response: Thanks for your comment. We have spelt properly the scientific name of B. subtilis as follows:
L(237-238)Some studies have shown that B. Subtilis has the ability to synthesize protease at a suitable pH of 7.0-7.4.
Comment 10. L(214-215). In the sentence: “The reason for the difference in bacteria density might be associated with the abundant nutrition in SMP comparing with other soy protein products. The data of bacteria density indicated that the growth preference of BSNK-5 was SMP”. More scientific evidence is needed to support this assertion. At the very least, it is suggested that the concept of "nutrition" be further analyzed. For example, to associate it with the concentration of proteins and the types of amino acids already analyzed above and to corroborate whether or not this has a significant influence. In addition, look for background information or reports that support this statement.
Author Response: Thanks for your kind comment. According to your comment, we have reviewed the relevant information. We have further analyzed the effect of SMP nutrients on the growth of BSNK-5 and looked for background information or reports that support this statement as follows:
L(244-252)The reason for the difference in bacteria density might be because of the abundant nutrition in SMP comparing with that in the other soy protein products. Other studies have found that SMP has more soluble proteins, and the growth of BSNK-5 will first consume the soluble proteins. Similarly, an appropriate carbon/nitrogen ratio has a great influence on the growth of B. Subtilis. The bacteria density of SMP was significantly higher than that of SPC, and the removal of carbohydrates would lead to the loss of nutrients in BSNK-5 growth. In addition to essential nutrients, some metal ions, such as Ca2+, K+ and Mn2+, promote the formation of bacteria and spores.
References
- Liang, Y.R.; Guo, Y.N.; Zheng, Y.X.; Liu, S.B.; Cheng, T.F.; Zhou, L.Y.; Guo, Z.W. Effects of high-pressure homogenization on physicochemical and functional properties of enzymatic hydrolyzed soybean protein concentrate. Front Nutr. 2022, 9, 1054326.
- Li, X.N. Optimization of cutlure conditions and probiotic potential evaluation of Bacillus Subtilis JB286. Shandong Agricultural University. 2023, 5.
Comment 11. L(227). In the sentence: “The NK produced by B. subtilis has potential in the treatment of cardiovascular diseases”. This information is relevant; however, it can go in the introduction so that it better justifies the work.
Author Response: Thanks for your kind reminder. We have added this part to the introduction and supplemented the application of the other two strains as follows:
L(43-54)Nattokinase (NK) and other functional food ingredients are produced during the fer-mentation of natto. NK, as an effective thrombolytic enzyme, has been intensively studied in clinical practice and is likely to become an important functional substance in the prevention of thrombotic cardiovascular diseases. Microorganisms also act as probiotic agents, for example, Lactobacillus, Bifidobacterium and Lactobacillus acidophilus, which regulate the intestinal ecosystem and improve the hosts’ health. Strep-tococcus lactis as a probiotic is characterized by the ability to use carbohydrates as substrates to produce lactic acid and nisin. Nisin, as an efficient, non-toxic, safe and non-side-effect-inducing natural food preservative additive, is added to dairy products, fermented beverages, canned foods and meat products to control the growth of pathogenic microorganisms such as Staphylococcus aureus and Listeria monocytogenes, thereby extending the shelf life of foods.
Comment 12. L(230). In the Figure 2.B, it is suggested to express CFU/mL values in another numerical scale (Ln or Log10). In addition, verify if the data are complete since the graph shows a cut-off. In general. Set up the graphs so that the variables tested can be differentiated. The curves appear very mixed between them and make a proper analysis difficult for the reader.
Author Response: We appreciate you very much for your comment. We have modified Figure 2 and expressed CFU/mL values in another numerical scale (Log10). The purpose of our graph is to reflect the trend of growth and metabolism indexes at different fermentation times, which would make the graph look less clear if we add significance indicators to it. We likewise did a significance analysis of different soy protein products as a source of nitrogen at the same point in time, which we can present in the form of a schedule if possible as follows:
Table 1 The pH (A), bacteria density (B) and NK activity (C) during the fermentation of five kinds of soy protein product.
|
Time(h) |
BE |
SP |
SMP |
SPC |
SPI |
SPH |
|
pH |
||||||
|
12 |
6.15 ±0.75a |
6.33 ±0.19a |
5.43 ±0.05a |
5.54 ±0.04a |
5.46 ±0.04a |
5.37 ±0.01a |
|
24 |
6.08 ±0.49a |
6.73 ±0.05a |
6.64 ±0.40a |
6.47 ±0.15a |
5.85 ±0.01a |
6.19 ±0.04a |
|
36 |
6.60 ±0.40a |
5.29 ±0.11b |
5.31 ±0.10b |
5.30 ±0.01b |
5.79 ±0.03b |
6.90 ±0.09a |
|
48 |
7.03 ±0.08a |
5.27 ±0.31cd |
5.27 ±0.05cd |
5.16 ±0.04d |
5.74 ±0.02bc |
6.18 ±0.04b |
|
60 |
5.88 ±0.46ab |
5.23 ±0.22b |
5.32 ±0.02ab |
5.19 ±0.02b |
5.89 ±0.06ab |
6.07 ±0.03a |
|
72 |
5.73 ±0.40ab |
5.06 ±0.05b |
5.31 ±0.01ab |
5.16 ±0.04b |
5.86 ±0.08a |
5.96 ±0.09a |
|
84 |
5.45 ±0.14b |
5.10 ±0.01c |
5.29 ±0.01bc |
5.16 ±0.01c |
6.14 ±0.02a |
6.19 ±0.03a |
|
96 |
5.51 ±0.17b |
5.08 ±0.01b |
5.29 ±0.00b |
5.15 ±0.04b |
6.34 ±0.20a |
6.30 ±0.21a |
|
108 |
5.67 ±0.14b |
5.12 ±0.00b |
5.30 ±0.04b |
5.14 ±0.04b |
8.27 ±0.16a |
7.97 ±0.63a |
|
120 |
5.39 ±0.00b |
5.04 ±0.02b |
5.30 ±0.01b |
5.18 ±0.02b |
8.78 ±0.08a |
8.72 ±0.21a |
|
Bacteria density |
||||||
|
12 |
8.72 ±0.11a |
7.74 ±0.41abc |
7.83 ±0.84ab |
8.33 ±0.10a |
6.83 ±0.12c |
7.32 ±0.15bc |
|
24 |
10.16 ±0.06a |
9.57 ±0.58a |
10.16 ±0.05a |
8.20 ±0.48b |
9.84 ±0.05a |
9.52 ±0.13a |
|
36 |
9.02 ±0.23a |
8.36 ±0.13bc |
7.07 ±0.22d |
8.63 ±0.08b |
7.00 ±0.12d |
8.05 ±0.12c |
|
48 |
8.89 ±0.20a |
7.04 ±0.14d |
7.59 ±0.09c |
8.37 ±0.04b |
7.64 ±0.11c |
8.80 ±0.17a |
|
60 |
8.98 ±0.08a |
7.17 ±0.11c |
7.46 ±0.33c |
8.08 ±0.22b |
7.60 ±0.10c |
8.87 ±0.18a |
|
72 |
8.67 ±0.28a |
6.64 ±0.14c |
7.57 ±0.04c |
8.22 ±0.11b |
7.64 ±0.11c |
8.90 ±0.21a |
|
84 |
8.82 ±0.24a |
6.77 ±0.15c |
7.62 ±0.42b |
8.50 ±0.18a |
7.59 ±0.08b |
8.92 ±0.11a |
|
96 |
8.51 ±0.21a |
6.96 ±0.03c |
7.78 ±0.49b |
8.39 ±0.05a |
7.63 ±0.11b |
8.90 ±0.14a |
|
108 |
8.49 ±0.02b |
6.88 ±0.05d |
7.55 ±0.10c |
8.36 ±0.04b |
8.21 ±0.26b |
9.00 ±0.12a |
|
120 |
8.35 ±0.01b |
6.86 ±0.02c |
7.33 ±0.04d |
8.37 ±0.07b |
8.00 ±0.06c |
8.89 ±0.15a |
|
NK activity |
||||||
|
12 |
1.58 ±0.38b |
2.41 ±0.07a |
1.52 ±0.06bc |
1.31 ±0.08c |
0.97 ±0.05d |
1.31 ±0.04c |
|
24 |
1.95 ±0.54b |
2.74 ±0.13a |
1.93 ±0.06b |
1.95 ±0.07b |
1.27 ±0.39c |
1.56 ±0.09bc |
|
36 |
2.58 ±0.31b |
3.24 ±0.12a |
2.32 ±0.18b |
2.26 ±0.17b |
1.29 ±0.11c |
2.09 ±0.06bc |
|
48 |
2.50 ±0.31a |
2.79 ±0.20a |
1.99 ±0.18b |
1.88 ±0.14b |
1.06 ±0.11c |
2.03 ±0.16b |
|
60 |
2.75±0.18a |
2.91 ±0.18a |
2.14 ±0.09b |
2.05 ±0.09b |
1.54 ±0.11c |
2.02 ±0.11b |
|
72 |
2.46 ±0.12ab |
2.48 ±0.06a |
1.98 ±0.08c |
1.84 ±0.09d |
1.23 ±0.08c |
2.35 ±0.08b |
|
84 |
2.58 ±0.28a |
2.47 ±0.16a |
1.86 ±0.05b |
1.90 ±0.09b |
1.46 ±0.05c |
2.45 ±0.09a |
|
96 |
2.69 ±0.25ab |
2.76 ±0.11a |
2.04 ±0.12c |
2.08 ±0.17c |
1.71 ±0.08d |
2.51 ±0.15b |
|
108 |
2.69 ±0.19a |
2.65 ±0.11a |
2.00 ±0.15b |
1.82 ±0.11bc |
1.69 ±0.15c |
2.66 ±0.14a |
|
120 |
2.89 ±0.28a |
2.78 ±0.11ab |
2.22 ±0.11c |
1.97 ±0.09d |
1.86 ±0.10c |
2.62 ±0.17b |
Comment 13. L(233). Proper spelling of the scientific name (S. lactis).
Author Response: Thanks for your comment. We have spelt properly the scientific name of S. lactis. as follows:
L(269-270)3.2.2. Effect of five kinds of soy protein product on cell growth and nisin synthesis of S. lactis.
Comment 14. L(257). In the Figure 3.B, it is suggested to express CFU/mL values in another numerical scale (Ln or Log10). In addition, verify if the data are complete since the graph shows a cut-off. In general. Set up the graphs so that the variables tested can be differentiated. The curves appear very mixed between them and make a proper analysis difficult for the reader.
Author Response: We appreciate you very much for your comment. We have modified Figure 3 and
expressed CFU/mL values in another numerical scale (Log10). The purpose of our graph is to reflect the trend of growth and metabolism indexes at different fermentation times, which would make the graph look less clear if we add significance indicators to it. We likewise did a significance analysis of different soy protein products as a source of nitrogen at the same point in time, which we can present in the form of a schedule if possible as follows:
Table 2 The pH (A), bacteria density (B) and nisin (C) during the fermentation of five kinds of soy protein product.
|
Time(h) |
BE |
SP |
SMP |
SPC |
SPI |
SPH |
|
pH |
||||||
|
4 |
5.57 ±0.01e |
6.51 ±0.01b |
6.12 ±0.03d |
6.71 ±0.01a |
6.57 ±0.02b |
6.40 ±0.03c |
|
8 |
5.08 ±0.01d |
5.73 ±0.03b |
5.42 ±0.02c |
6.56 ±0.02a |
5.73 ±0.06b |
5.46 ±0.00c |
|
12 |
4.97 ±0.00d |
5.39 ±0.02b |
5.18 ±0.01c |
6.36 ±0.01a |
5.35 ±0.04b |
5.21 ±0.01c |
|
16 |
3.94 ±0.00d |
4.33 ±0.01b |
4.12 ±0.01c |
5.34 ±0.01a |
4.29 ±0.05b |
4.25 ±0.05b |
|
20 |
3.90 ±0.01d |
4.25 ±0.01b |
4.04 ±0.01c |
5.25 ±0.04a |
4.20 ±0.06b |
4.12 ±0.03bc |
|
24 |
3.90 ±0.01d |
4.21 ±0.02b |
4.03 ±0.01d |
5.22 ±0.04a |
4.18 ±0.05b |
4.11 ±0.00bc |
|
Bacteria density |
||||||
|
4 |
8.68 ±0.17a |
8.24 ±0.08bc |
8.11 ±0.06bc |
8.44 ±0.29ab |
8.72 ±0.12a |
8.04 ±0.04c |
|
8 |
8.82 ±0.17abc |
8.70 ±0.00bc |
8.62 ±0.15c |
8.96 ±0.10ab |
9.01 ±0.04a |
8.25 ±0.14d |
|
12 |
8.73 ±0.05a |
8.79 ±0.08a |
8.59 ±0.11a |
8.62 ±0.22a |
8.79 ±0.08a |
8.16 ±0.08b |
|
16 |
8.69 ±0.19a |
8.41 ±0.06ab |
8.42 ±0.11ab |
8.65 ±0.17a |
8.78 ±0.25a |
8.13 ±0.18b |
|
20 |
10.36 ±0.08a |
9.46 ±0.03b |
8.63 ±0.06c |
8.76 ±0.13c |
8.71 ±0.02c |
8.24 ±0.02d |
|
24 |
9.88 ±0.02a |
8.42 ±0.06c |
8.40 ±0.11c |
8.65 ±0.07bc |
8.82 ±0.04b |
8.03 ±0.11d |
|
Nisin |
||||||
|
4 |
20.69 ±7.20ab |
36.14 ±20.72a |
15.55 ±6.55b |
6.74 ±1.46b |
13.42 ±2.50b |
12.92 ±2.77b |
|
8 |
31.09 ±10.83bc |
36.44 ±11.25ab |
21.24 ±10.67bc |
25.33 ±10.21bc |
52.94 ±21.21a |
13.46 ±3.56c |
|
12 |
42.53 ±19.88a |
22.44 ±4.24b |
9.77 ±1.96b |
8.89 ±3.56b |
13.61 ±9.53b |
20.97 ±12.29b |
|
16 |
16.29 ±2.75bc |
22.89 ±7.63ab |
12.90 ±2.99bc |
23.88 ±11.29abc |
35.32 ±14.96a |
12.03 ±7.18c |
|
20 |
19.13 ±0.75bc |
56.56 ±12.55a |
32.83 ±25.52b |
27.71 ±8.85bc |
11.32 ±5.08c |
19.56 ±8.45bc |
|
24 |
9.87 ±2.84ab |
8.72 ±3.74ab |
4.57 ±2.44b |
8.58 ±2.55ab |
9.41 ±7.32ab |
16.64 ±10.36a |
Comment 15. L(260). Proper spelling of the scientific name (S. clavuligerus).
Author Response: Thanks for your comment. We have spelt properly the scientific name of S. clavuligerus as follows:
L(299-300)3.2.3. Effect of five kinds of soy protein product on cell growth and CA synthesis of S. clavuligerus.
Comment 16. L(281). In the Figure 4. In general. Set up the graphs so that the variables tested can be differentiated. The curves appear very mixed between them and make a proper analysis difficult for the reader.
Author Response: We appreciate you very much for your comment. We have modified Figure 4 and expressed CFU/mL values in another numerical scale (Log10). The purpose of our graph is to reflect the trend of growth and metabolism indexes at different fermentation times, which would make the graph look less clear if we add significance indicators to it. We likewise did a significance analysis of different soy protein products as a source of nitrogen at the same point in time, which we can present in the form of a schedule if possible as follows:
Table 3 The pH (A), PMV (B) and CA (C) during the fermentation of five kinds of soy protein product.
|
Time(h) |
BE |
SP |
SMP |
SPC |
SPI |
SPH |
|
pH |
||||||
|
24 |
7.02 ±0.01c |
7.11 ±0.01ab |
7.07 ±0.01bc |
7.11 ±0.01ab |
7.17 ±0.01a |
7.12 ±0.04ab |
|
48 |
7.03 ±0.01ab |
7.14 ±0.01ab |
7.09 ±0.01ab |
7.19 ±0.00a |
7.14 ±0.00ab |
6.91 ±0.16b |
|
72 |
6.95 ±0.03a |
6.61 ±0.06b |
7.02 ±0.01a |
6.97 ±0.01a |
6.96 ±0.04a |
6.94 ±0.00a |
|
96 |
7.32 ±0.04ab |
7.41 ±0.03ab |
7.42 ±0.01a |
7.30 ±0.03b |
7.40 ±0.01ab |
7.33 ±0.04ab |
|
120 |
7.09 ±0.08b |
7.06 ±0.04b |
7.36 ±0.07ab |
7.07 ±0.18b |
7.46 ±0.09ab |
7.51 ±0.08a |
|
144 |
6.97 ±0.11b |
7.23 ±0.21ab |
7.88 ±0.22a |
7.09 ±0.01b |
7.54 ±0.25ab |
7.90 ±0.07a |
|
PMV |
||||||
|
24 |
5.00 ±0.01d |
2.00 ±0.01d |
13.00 ±0.01c |
24.00 ±0.00a |
5.00 ±0.01d |
18.00 ±0.00b |
|
48 |
9.00 ±0.04c |
7.00 ±0.01c |
31.00 ±0.04a |
26.00 ±0.00ab |
8.00 ±0.01c |
16.00 ±0.00bc |
|
72 |
3.00 ±0.01c |
14.00 ±0.00b |
37.00 ±0.02a |
33.00 ±0.02a |
37.00 ±0.04a |
19.00 ±0.01b |
|
96 |
20.00 ±0.00a |
21.00 ±0.01a |
24.00 ±0.16a |
33.00 ±0.03a |
21.00 ±0.01a |
9.00 ±0.04a |
|
120 |
49.00 ±0.01a |
24.00 ±0.03b |
31.00 ±0.01ab |
34.00±0.09ab |
47.00 ±0.04a |
35.00 ±0.07ab |
|
144 |
39.00 ±0.01a |
18.00 ±0.00d |
28.00 ±0.03bc |
35.00 ±0.01ab |
23.00 ±0.04cd |
16.00 ±0.01d |
|
CA |
||||||
|
24 |
2.98 ±0.11c |
3.03 ±0.04c |
3.38 ±0.07ab |
3.46 ±0.06a |
3.03 ±0.04c |
3.28 ±0.09b |
|
48 |
3.28 ±0.09ab |
3.05 ±0.05c |
3.46 ±0.12a |
3.40 ±0.07a |
3.10 ±0.08bc |
3.46 ±0.14a |
|
72 |
3.23 ±0.11c |
3.67 ±0.00ab |
3.83 ±0.04a |
3.73 ±0.01ab |
3.47 ±0.35bc |
3.64 ±0.06ab |
|
96 |
3.49 ±0.03bc |
3.43 ±0.29c |
3.81 ±0.05a |
3.74 ±0.01ab |
3.68 ±0.04abc |
3.71 ±0.03ab |
|
120 |
3.48 ±0.06bc |
3.41 ±0.21c |
3.84 ±0.03a |
3.63 ±0.14abc |
3.72 ±0.04a |
3.70 ±0.02ab |
|
144 |
3.40 ±0.03d |
3.50 ±0.20cd |
3.74 ±0.02ab |
3.81 ±0.01a |
3.68 ±0.02abc |
3.61 ±0.02bc |
Comment 17. In conclusion, according to the results shown, I consider that a soy protein source suitable for the growth of the 3 microorganisms worked on should be chosen, depending on the biomass production yield or cell density and the metabolites of each one, or failing that, choose one for each microorganism. Likewise, it is also worth mentioning to show in this section, the most outstanding quantitative values obtained according to their objectives.
Author Response: Thanks for your kind reminder. We have modified the conclusion according to your opinion. We have showed that a soy protein source suitable for the growth of the three strains worked on should be chosen, depending on the biomass production yield or cell density and the metabolites of each one. The most outstanding quantitative values obtained according to their objectives were showed in this section as follows:
L(393-414)Based on the rapid growth of the microbiology industry and high-quality nutritional components of soy protein products, it is necessary to promote cell growth and metabolism at a low cost in order to develop widely applicable medium nitrogen sources. The total nitrogen contents and protein concentrations of the five kinds of soy protein product were significantly different, and the total nitrogen content of each sample was always higher than its protein content. The total nitrogen contents and protein concentrations of SPI and SPH were both higher than those of SP, SMP and SPC. The contents of 17 amino acids in the five kinds of soy protein product were ordered as follows: SPC > SPI > SMP > SPH > SP. Among them, the contents of Glu, Asp, Leu and Arg were higher, and the contents of Met and Cys were lower. When SMP was used as a nitrogen source, BSNK-5 had the highest bacteria density (4.67×1010 CFU/mL). SP had the highest NK activity level (3.2×104 IU/mL), which was higher than the NK activity levels of the other soy protein products by 28%-60%. When SPI was used as a nitrogen source, S. lactis had the highest bacteria density (1.0×109 CFU/mL), and nisin had the highest inhibitory activity level, which was 45%-149% higher than the inhibitory activities of the other soy protein products. SPI as a nitrogen source showed the highest PMV (50%). SMP as a nitrogen source showed the highest CA content (6852.5 mg/L), which was 28%-60% higher than the CA contents of the other soy protein products. On this basis, the correlation analysis of the fermentation broth components, the cell growth and the metabolism of the strains showed that the pH and amino acids were closely related to the production of NK, nisin and CA. Taken together, these findings suggest that soy protein products have the potential to be novel microbial nitrogen sources.

Reviewer 2 Report
Comments and Suggestions for Authors
The present work is relevant to find sustainable and economical ways to obtain bacterial biomass and their respective bioactive products.
However, several points must be improved prior to the publication of this manuscript.
First of all, it would be necessary to add the differences between the forms of soy proteins used, their source of obtaining and/or processing.
As well as adding in the conclusions how the type of products is related to the composition of amino acids.
In relation to microorganisms and their growth conditions: although the information is complete, it is not adequately described and is a bit confusing to read, which is why it is recommended to rewrite this section.
Regarding the growth curves (pH and microbial density), as well as the production of bioactive compounds, it is necessary to add in the graph and in the text the statistical parameters that justify the differences mentioned between the different protein sources used (level of significance) and check that there are no conditions that do not show significant differences, which is why they would be equivalent when formulating a culture medium with these alternative nitrogen sources.
Author Response
Reviewer #2:
Comment 1. The present work is relevant to find sustainable and economical ways to obtain bacterial biomass and their respective bioactive products.
However, several points must be improved prior to the publication of this manuscript.
(1) First of all, it would be necessary to add the differences between the forms of soy proteins used, their source of obtaining and/or processing.
(2) As well as adding in the conclusions how the type of products is related to the composition of amino acids.
(3) In relation to microorganisms and their growth conditions: although the information is complete, it is not adequately described and is a bit confusing to read, which is why it is recommended to rewrite this section.
(4) Regarding the growth curves (pH and microbial density), as well as the production of bioactive compounds, it is necessary to add in the graph and in the text the statistical parameters that justify the differences mentioned between the different protein sources used (level of significance) and check that there are no conditions that do not show significant differences, which is why they would be equivalent when formulating a culture medium with these alternative nitrogen sources.
Author Response: Thanks for your kind reminder. We supplemented the differences in protein content, source of acquisition and processing technology of five soy protein products, rewritten the microorganisms and their growth conditions, and added the relationship between the type of soybean protein products and the amino acid composition in the conclusions. The purpose of our graph is to reflect the trend of growth metabolism indicators at different fermentation times, and adding significance indicators to the graph would make it look less clear. We likewise did a significance analysis of different soy protein products as nitrogen sources at the same point in time, which we can present in the form of a table if possible.
(1)L(73-93)Soybean meal is a by-product of soybean oil extraction from soybeans and has a protein content of about 44.0%-53.1%. Due to its high protein content, balanced amino acid composition and abundant supply, it is a major source of high-quality protein. SPC is obtained by removing the carbohydrates and lipids from soybean meal and has a protein content of more than 70%. There are two methods of preparation: (1) Acid leaching involves the precipitation of water-insoluble fibers and proteins using isoelectric point precipitation. (2) Alcohol leaching comprises dissolving soluble carbohydrates and alcohol-soluble polyphenols, leaving the soy proteins. SPC is considered to have a high nutritional value because it is rich in essential fatty acids, phospholipids, minerals, and nine essential amino acids, and has a balanced amino acid composition similar to that of animal proteins and a digestibility of up to 90%. SPI is obtained by dissolving the proteins from soybean meal at a high pH, removing the insoluble fibers using centrifugation to obtain a supernatant, and reacquiring the proteins at the isoelectric point, with a content of about 85%-90%. SPH is a mixture of soy proteins that have been converted into oligopeptides, polypeptides and amino acids prepared either using protein processing processes, such as hydrolysis and a heat treatment, or biological processes, including digestion and microbial fermentation. SP is a powdered product prepared from soybean meal using enzymatic hydrolysis, filtration, concentration and spray drying. SP contains proteins, peptides, amino acids, water-soluble vitamins and many sugars. Soluble carbohydrates mainly include sucrose, stachyose, raffinose and verbascose, which function as gene inducers in specific expression systems.
(2)L(399-402)The contents of 17 amino acids in the five kinds of soy protein product were ordered as follows: SPC > SPI > SMP > SPH > SP. Among them, the contents of Glu, Asp, Leu and Arg were higher, and the contents of Met and Cys were lower.
(3)L(328-389)As shown in Figure 5, blue indicates a positive correlation, while red indicates the opposite. A positive correlation indicates a promoting effect, and a negative correlation indicates an inhibiting effect. Figure 5 revealed the correlation between the nutritional components of soy protein products, the cell growth of BSNK-5, and NK synthesis. It has been reported that the pH influences bacterial growth and enzyme synthesis. There was a negative correlation between the pH and bacteria density. NK was synthesized by BSNK-5, and the fermentation broth environment was weakly alkaline, which made the activity level of NK activity as an alkaline serine enzyme higher. Since BSNK-5 utilized glucose to produce organic acids, resulting in a decrease in the pH of the fermentation broth, correlation analysis validated a negative correlation between the pH and NK. At 36 h of fermentation, the bacteria density was positively correlated with the peptides and Pro. In addition, the NK activity was positively correlated with the peptides and Pro. Bacteria density was positively correlated with NK. Other studies have shown that NK synthesis consumes energy, and the energy source is mainly amino acids. The higher the degree of hydrolysis of soybean protein is, the higher the concentrations of peptides and amino acids are. NK synthesis consumes peptides, resulting in an increase in NK production.
Figure 6 indicated the nutritional components of the soy protein products, the cell growth of S. lactis, and nisin synthesis. The bacteria density was positively correlated with the protein content, but when the substrate concentration was too high, the growth of the bacteria was inhibited. The bacteria density was negatively correlated with nisin. With the growth of S. lactis, nisin was continuously produced. The reason for the negative correlation between bacteria density and nisin might be that the strain entered a stable period and nisin was no longer produced. The production of nisin was heavily dependent on the biomass of the strain, and high-density strains promoted the production of nisin. Therefore, many studies aim to increase nisin production by increasing the physiological activity of bacteria or the number of active nisin-producing bacteria during fermentation. One of the most basic and effective methods is to increase the bacteria density. Bacteria density was negatively correlated with the pH, and a lower pH inhibited the growth of bacteria. Nisin was positively correlated with the pH, and the accumulation of lactic acid led to a decrease in the pH, which, in turn, inhibited the growth of S. lactis. Nisin is highly stable at a low pH. With an increase in pH, the stability of nisin decreases greatly. After heating at 121°C for 15 min, the nisin activity level decreased by 29% at pH 4.0, 69% at pH 5.0 and 86% at pH 6.0. When the pH changed to 7.0, 99.7% of the nisin lost its antibacterial activity. Nisin was positively correlated with Gly, Ala, Met, Glu, Ser and Pro. Other studies have shown that adding the precursor amino acids of nisin (Cys, Thr and Ser) and sulfur-containing amino acids (Met and Cys) to a basic medium can increase the titer of nisin in the fermentation broth. Therefore, the concentration of nisin can be increased by modulating the pH environment and amino acid composition of the medium.
The correlation between the nutritional components of soy protein products, the cell growth of S. clavuligerus and CA synthesis was shown in Figure 7. The protein content was closely related to the cell growth and CA content of S. clavuligerus. PMV was positively correlated with Leu, Val, Ile, Asp and Glu. PMV was negatively correlated with Lys, Arg, Gly, Ala and Pro. CA was positively correlated with Met, Pro, Lys, Gly, Ala and Arg. Some experiments have shown that the addition of certain amino acids (Thr, Glu and Asp) does not aid or even inhibit CA synthesis. Complex medium that provided large amounts of hydrolyzed or free amino acids are more favorable for CA production than chemically defined medium. For example, Lys, Val, Tyr, Phe and Cys have strong inhibitory effects. Rodrigues found that by adding different concentrations of Arg, Thr, Orn and Glu to a fermentation medium in which soy isolate proteins were used as a nitrogen source, Arg and Thr had no effect on CA production, while the addition of the other two amino acids inhibited CA production. The current findings on CA synthesis are inconsistent, with the addition of Arg and Thr favoring CA synthesis, which may be due to the inconsistency of the strains. In response to these results, the CA concentration could be increased by the addition of amino acid supplements to provide C-5 precursors for the urea cycle.
(4)The significance analysis is shown in the table below, but due to the large amount of data in the growth and metabolism graphs, it was not possible to label the graphs, so significance has been added to the right side of the graphs. We can add schedules if they need to be presented in the following table.
Figure 2.
Table 1 The pH (A), bacteria density (B) and NK activity (C) during the fermentation of five kinds of soy protein product.
|
Time(h) |
BE |
SP |
SMP |
SPC |
SPI |
SPH |
|
pH |
||||||
|
12 |
6.15 ±0.75a |
6.33 ±0.19a |
5.43 ±0.05a |
5.54 ±0.04a |
5.46 ±0.04a |
5.37 ±0.01a |
|
24 |
6.08 ±0.49a |
6.73 ±0.05a |
6.64 ±0.40a |
6.47 ±0.15a |
5.85 ±0.01a |
6.19 ±0.04a |
|
36 |
6.60 ±0.40a |
5.29 ±0.11b |
5.31 ±0.10b |
5.30 ±0.01b |
5.79 ±0.03b |
6.90 ±0.09a |
|
48 |
7.03 ±0.08a |
5.27 ±0.31cd |
5.27 ±0.05cd |
5.16 ±0.04d |
5.74 ±0.02bc |
6.18 ±0.04b |
|
60 |
5.88 ±0.46ab |
5.23 ±0.22b |
5.32 ±0.02ab |
5.19 ±0.02b |
5.89 ±0.06ab |
6.07 ±0.03a |
|
72 |
5.73 ±0.40ab |
5.06 ±0.05b |
5.31 ±0.01ab |
5.16 ±0.04b |
5.86 ±0.08a |
5.96 ±0.09a |
|
84 |
5.45 ±0.14b |
5.10 ±0.01c |
5.29 ±0.01bc |
5.16 ±0.01c |
6.14 ±0.02a |
6.19 ±0.03a |
|
96 |
5.51 ±0.17b |
5.08 ±0.01b |
5.29 ±0.00b |
5.15 ±0.04b |
6.34 ±0.20a |
6.30 ±0.21a |
|
108 |
5.67 ±0.14b |
5.12 ±0.00b |
5.30 ±0.04b |
5.14 ±0.04b |
8.27 ±0.16a |
7.97 ±0.63a |
|
120 |
5.39 ±0.00b |
5.04 ±0.02b |
5.30 ±0.01b |
5.18 ±0.02b |
8.78 ±0.08a |
8.72 ±0.21a |
|
Bacteria density |
||||||
|
12 |
8.72 ±0.11a |
7.74 ±0.41abc |
7.83 ±0.84ab |
8.33 ±0.10a |
6.83 ±0.12c |
7.32 ±0.15bc |
|
24 |
10.16 ±0.06a |
9.57 ±0.58a |
10.16 ±0.05a |
8.20 ±0.48b |
9.84 ±0.05a |
9.52 ±0.13a |
|
36 |
9.02 ±0.23a |
8.36 ±0.13bc |
7.07 ±0.22d |
8.63 ±0.08b |
7.00 ±0.12d |
8.05 ±0.12c |
|
48 |
8.89 ±0.20a |
7.04 ±0.14d |
7.59 ±0.09c |
8.37 ±0.04b |
7.64 ±0.11c |
8.80 ±0.17a |
|
60 |
8.98 ±0.08a |
7.17 ±0.11c |
7.46 ±0.33c |
8.08 ±0.22b |
7.60 ±0.10c |
8.87 ±0.18a |
|
72 |
8.67 ±0.28a |
6.64 ±0.14c |
7.57 ±0.04c |
8.22 ±0.11b |
7.64 ±0.11c |
8.90 ±0.21a |
|
84 |
8.82 ±0.24a |
6.77 ±0.15c |
7.62 ±0.42b |
8.50 ±0.18a |
7.59 ±0.08b |
8.92 ±0.11a |
|
96 |
8.51 ±0.21a |
6.96 ±0.03c |
7.78 ±0.49b |
8.39 ±0.05a |
7.63 ±0.11b |
8.90 ±0.14a |
|
108 |
8.49 ±0.02b |
6.88 ±0.05d |
7.55 ±0.10c |
8.36 ±0.04b |
8.21 ±0.26b |
9.00 ±0.12a |
|
120 |
8.35 ±0.01b |
6.86 ±0.02c |
7.33 ±0.04d |
8.37 ±0.07b |
8.00 ±0.06c |
8.89 ±0.15a |
|
NK activity |
||||||
|
12 |
1.58 ±0.38b |
2.41 ±0.07a |
1.52 ±0.06bc |
1.31 ±0.08c |
0.97 ±0.05d |
1.31 ±0.04c |
|
24 |
1.95 ±0.54b |
2.74 ±0.13a |
1.93 ±0.06b |
1.95 ±0.07b |
1.27 ±0.39c |
1.56 ±0.09bc |
|
36 |
2.58 ±0.31b |
3.24 ±0.12a |
2.32 ±0.18b |
2.26 ±0.17b |
1.29 ±0.11c |
2.09 ±0.06bc |
|
48 |
2.50 ±0.31a |
2.79 ±0.20a |
1.99 ±0.18b |
1.88 ±0.14b |
1.06 ±0.11c |
2.03 ±0.16b |
|
60 |
2.75±0.18a |
2.91 ±0.18a |
2.14 ±0.09b |
2.05 ±0.09b |
1.54 ±0.11c |
2.02 ±0.11b |
|
72 |
2.46 ±0.12ab |
2.48 ±0.06a |
1.98 ±0.08c |
1.84 ±0.09d |
1.23 ±0.08c |
2.35 ±0.08b |
|
84 |
2.58 ±0.28a |
2.47 ±0.16a |
1.86 ±0.05b |
1.90 ±0.09b |
1.46 ±0.05c |
2.45 ±0.09a |
|
96 |
2.69 ±0.25ab |
2.76 ±0.11a |
2.04 ±0.12c |
2.08 ±0.17c |
1.71 ±0.08d |
2.51 ±0.15b |
|
108 |
2.69 ±0.19a |
2.65 ±0.11a |
2.00 ±0.15b |
1.82 ±0.11bc |
1.69 ±0.15c |
2.66 ±0.14a |
|
120 |
2.89 ±0.28a |
2.78 ±0.11ab |
2.22 ±0.11c |
1.97 ±0.09d |
1.86 ±0.10c |
2.62 ±0.17b |
Figure 3.
|
Time(h) |
BE |
SP |
SMP |
SPC |
SPI |
SPH |
|
pH |
||||||
|
4 |
5.57 ±0.01e |
6.51 ±0.01b |
6.12 ±0.03d |
6.71 ±0.01a |
6.57 ±0.02b |
6.40 ±0.03c |
|
8 |
5.08 ±0.01d |
5.73 ±0.03b |
5.42 ±0.02c |
6.56 ±0.02a |
5.73 ±0.06b |
5.46 ±0.00c |
|
12 |
4.97 ±0.00d |
5.39 ±0.02b |
5.18 ±0.01c |
6.36 ±0.01a |
5.35 ±0.04b |
5.21 ±0.01c |
|
16 |
3.94 ±0.00d |
4.33 ±0.01b |
4.12 ±0.01c |
5.34 ±0.01a |
4.29 ±0.05b |
4.25 ±0.05b |
|
20 |
3.90 ±0.01d |
4.25 ±0.01b |
4.04 ±0.01c |
5.25 ±0.04a |
4.20 ±0.06b |
4.12 ±0.03bc |
|
24 |
3.90 ±0.01d |
4.21 ±0.02b |
4.03 ±0.01d |
5.22 ±0.04a |
4.18 ±0.05b |
4.11 ±0.00bc |
|
Bacteria density |
||||||
|
4 |
8.68 ±0.17a |
8.24 ±0.08bc |
8.11 ±0.06bc |
8.44 ±0.29ab |
8.72 ±0.12a |
8.04 ±0.04c |
|
8 |
8.82 ±0.17abc |
8.70 ±0.00bc |
8.62 ±0.15c |
8.96 ±0.10ab |
9.01 ±0.04a |
8.25 ±0.14d |
|
12 |
8.73 ±0.05a |
8.79 ±0.08a |
8.59 ±0.11a |
8.62 ±0.22a |
8.79 ±0.08a |
8.16 ±0.08b |
|
16 |
8.69 ±0.19a |
8.41 ±0.06ab |
8.42 ±0.11ab |
8.65 ±0.17a |
8.78 ±0.25a |
8.13 ±0.18b |
|
20 |
10.36 ±0.08a |
9.46 ±0.03b |
8.63 ±0.06c |
8.76 ±0.13c |
8.71 ±0.02c |
8.24 ±0.02d |
|
24 |
9.88 ±0.02a |
8.42 ±0.06c |
8.40 ±0.11c |
8.65 ±0.07bc |
8.82 ±0.04b |
8.03 ±0.11d |
|
Nisin |
||||||
|
4 |
20.69 ±7.20ab |
36.14 ±20.72a |
15.55 ±6.55b |
6.74 ±1.46b |
13.42 ±2.50b |
12.92 ±2.77b |
|
8 |
31.09 ±10.83bc |
36.44 ±11.25ab |
21.24 ±10.67bc |
25.33 ±10.21bc |
52.94 ±21.21a |
13.46 ±3.56c |
|
12 |
42.53 ±19.88a |
22.44 ±4.24b |
9.77 ±1.96b |
8.89 ±3.56b |
13.61 ±9.53b |
20.97 ±12.29b |
|
16 |
16.29 ±2.75bc |
22.89 ±7.63ab |
12.90 ±2.99bc |
23.88 ±11.29abc |
35.32 ±14.96a |
12.03 ±7.18c |
|
20 |
19.13 ±0.75bc |
56.56 ±12.55a |
32.83 ±25.52b |
27.71 ±8.85bc |
11.32 ±5.08c |
19.56 ±8.45bc |
|
24 |
9.87 ±2.84ab |
8.72 ±3.74ab |
4.57 ±2.44b |
8.58 ±2.55ab |
9.41 ±7.32ab |
16.64 ±10.36a |
Table 2 The pH (A), bacteria density (B) and nisin (C) during the fermentation of five kinds of soy protein product.
Figure 4.
Table 3 The pH (A), PMV (B) and CA (C) during the fermentation of five kinds of soy protein product.
|
Time(h) |
BE |
SP |
SMP |
SPC |
SPI |
SPH |
|
pH |
||||||
|
24 |
7.02 ±0.01c |
7.11 ±0.01ab |
7.07 ±0.01bc |
7.11 ±0.01ab |
7.17 ±0.01a |
7.12 ±0.04ab |
|
48 |
7.03 ±0.01ab |
7.14 ±0.01ab |
7.09 ±0.01ab |
7.19 ±0.00a |
7.14 ±0.00ab |
6.91 ±0.16b |
|
72 |
6.95 ±0.03a |
6.61 ±0.06b |
7.02 ±0.01a |
6.97 ±0.01a |
6.96 ±0.04a |
6.94 ±0.00a |
|
96 |
7.32 ±0.04ab |
7.41 ±0.03ab |
7.42 ±0.01a |
7.30 ±0.03b |
7.40 ±0.01ab |
7.33 ±0.04ab |
|
120 |
7.09 ±0.08b |
7.06 ±0.04b |
7.36 ±0.07ab |
7.07 ±0.18b |
7.46 ±0.09ab |
7.51 ±0.08a |
|
144 |
6.97 ±0.11b |
7.23 ±0.21ab |
7.88 ±0.22a |
7.09 ±0.01b |
7.54 ±0.25ab |
7.90 ±0.07a |
|
PMV |
||||||
|
24 |
5.00 ±0.01d |
2.00 ±0.01d |
13.00 ±0.01c |
24.00 ±0.00a |
5.00 ±0.01d |
18.00 ±0.00b |
|
48 |
9.00 ±0.04c |
7.00 ±0.01c |
31.00 ±0.04a |
26.00 ±0.00ab |
8.00 ±0.01c |
16.00 ±0.00bc |
|
72 |
3.00 ±0.01c |
14.00 ±0.00b |
37.00 ±0.02a |
33.00 ±0.02a |
37.00 ±0.04a |
19.00 ±0.01b |
|
96 |
20.00 ±0.00a |
21.00 ±0.01a |
24.00 ±0.16a |
33.00 ±0.03a |
21.00 ±0.01a |
9.00 ±0.04a |
|
120 |
49.00 ±0.01a |
24.00 ±0.03b |
31.00 ±0.01ab |
34.00±0.09ab |
47.00 ±0.04a |
35.00 ±0.07ab |
|
144 |
39.00 ±0.01a |
18.00 ±0.00d |
28.00 ±0.03bc |
35.00 ±0.01ab |
23.00 ±0.04cd |
16.00 ±0.01d |
|
CA |
||||||
|
24 |
2.98 ±0.11c |
3.03 ±0.04c |
3.38 ±0.07ab |
3.46 ±0.06a |
3.03 ±0.04c |
3.28 ±0.09b |
|
48 |
3.28 ±0.09ab |
3.05 ±0.05c |
3.46 ±0.12a |
3.40 ±0.07a |
3.10 ±0.08bc |
3.46 ±0.14a |
|
72 |
3.23 ±0.11c |
3.67 ±0.00ab |
3.83 ±0.04a |
3.73 ±0.01ab |
3.47 ±0.35bc |
3.64 ±0.06ab |
|
96 |
3.49 ±0.03bc |
3.43 ±0.29c |
3.81 ±0.05a |
3.74 ±0.01ab |
3.68 ±0.04abc |
3.71 ±0.03ab |
|
120 |
3.48 ±0.06bc |
3.41 ±0.21c |
3.84 ±0.03a |
3.63 ±0.14abc |
3.72 ±0.04a |
3.70 ±0.02ab |
|
144 |
3.40 ±0.03d |
3.50 ±0.20cd |
3.74 ±0.02ab |
3.81 ±0.01a |
3.68 ±0.02abc |
3.61 ±0.02bc |

Round 2
Reviewer 1 Report
Comments and Suggestions for Authors
The article entitled “Effect of Soy Protein Products on The Growth and Metabolism of Bacillus subtilis Streptococcus lactis and Streptomyces clavuligerus” has been significantly improved, however I think some things still need to be adjusted in the manuscript.
I think this keyword “Correlation” should be removed
The quality of the figures is still poor
The conclusion needs to be redone into another abstract
Author Response
Reviewer #1:
Comment 1. I think this keyword “Correlation” should be removed. The quality of the figures is still poor. The conclusion needs to be redone into another abstract.
Author Response: Thanks for your nice comments. We have removed the keyword “Correlation” and improved the quality of the figures. We have rewritten the introduction and conclusion sections as follows:
L(14-34)Abstract:
Microbial nitrogen sources are promising, and soy protein as a plant-based nitrogen source has absolute advantages to create microbial culture medium in terms of renewability, eco-friendliness, and greater safety. Soy protein was rich in variety due to the different extraction technology and significantly different in cell growth and metabolism of microorganisms as nitrogen source. Therefore, the different soy proteins (soy meal powder, SMP; soy peptone, SP; soy protein concen-trate, SPC; soy protein isolate, SPI; soy protein hydrolysate, SPH) were used as nitrogen sources to culture Bacillus subtilis, Streptococcus lactis and Streptomyces clavuligerus for evaluating the suitable soy nitrogen source of the above strains. The results showed that B. subtilis had the highest bacteria density in SMP medium; S. lactis had the highest bacteria density in SPI medium; and S. clavuligerus had the highest PMV in SPI medium. Nattokinase activity was the highest in SP medium; bacteriostatic effect of nisin was the best in SPI medium; and clavulanic acid concentration was the highest in SMP medium. Based on analyzing the correlation between nutritional composition and growth metabolism of the strains, the results indicated that the protein content and amino acid composition were the key factors of influencing the cell growth and metabolism of the strains. These findings present a new, high-value application opportunity for soybean protein.
L(382-408)Conclusions:
Based on the rapid growth of the microbiology industry and high-quality nutritional components of soy protein products, it is necessary to promote cell growth and metabolism at a low cost in order to develop widely applicable medium nitrogen sources. The total nitrogen contents and protein concentrations of the five kinds of soy protein product were significantly different, and the total nitrogen content of each sample was always higher than its protein content. The total nitrogen contents and protein concentrations of SPI and SPH were both higher than those of SP, SMP and SPC. The contents of 17 amino acids in the five kinds of soy protein product were ordered as follows: SPC > SPI > SMP > SPH > SP. Among them, the contents of Glu, Asp, Leu and Arg were higher, and the contents of Met and Cys were lower. BSNK-5 had the highest bacteria density in SMP medium and. had the highest NK activity level in SP medium. S. lactis had the highest bacteria density in SPI medium and nisin had the highest inhibitory activity level. S. clavuligerus had the highest PMV in SPI medium and had the highest CA content in SMP medium. The correlation analysis of the nutritional composition in fermentation broth, the cell growth and the metabolism of the strains were closely related to the protein content and amino acids. The bacteria den-sity of BSNK-5 and NK activity were positively correlated with the peptide content. The bacteria density of S. lactis was positively correlated with the protein content. Nisin was positively correlated with certain amino acids. PMV of S. clavuligerus and CA content were both positively correlated with protein content. These findings suggest that soy protein products have the potential to be novel microbial nitrogen sources.

Reviewer 2 Report
Comments and Suggestions for Authors
The revised manuscript has been very satisfactorily improved
Minnor comment
In the introduccion, as mention of probiotics microorganism deleted Lactobacillus acidophilus, and only mention the genera Lactobacillus and Bifidobacterium
Line: 230, check the unit: mg of aminoacids per g or mg of sample???
Author Response
Reviewer #2:
Comment 1. In the introduction, as mention of probiotics microorganism deleted Lactobacillus acidophilus, and only mention the genera Lactobacillus and Bifidobacterium
Line: 230, check the unit: mg of amino acids per g or mg of sample???
Author Response: We appreciate you very much for your comment. L(46-47)We have deleted Lactobacillus acidophilus. L(226-227)The unit have been modified to mg of amino acids per g of the samples.
